# Case Study: Modeling a Grain Bin for Safe Entry Retrofit †

Michael Dyer [1,2], Serap Gorucu [3] , Randall Bock [1], Roderick Thomas [1], Jude Liu [1] and Linda Fetzer [1,*]

1. Department of Agricultural & Biological Engineering, Pennsylvania State University, University Park, PA 16801, USA; michael.dyer@usu.edu (M.D.); rgb@psu.edu (R.B.); rst5@psu.edu (R.T.); jliu@psu.edu (J.L.)
2. Environmental Health and Safety, Office of Research, Utah State University, Logan, UT 84322, USA
3. Department of Agricultural & Biological Engineering, University of Florida, Gainesville, FL 32611, USA; serapgorucu@ufl.edu
* Correspondence: lmf8@psu.edu
† This paper is an extended version of our conference paper: Gorucu, S.; Dyer, M.; Bock, R.; Thomas, R.; Fetzer, L.; Brown, S. Modeling a Grain Bin for Safe Entry Anchor Point Retrofit. ASABE Annual International Meeting, American Society of Agricultural and Biological Engineers, St. Joseph, MI, USA, July 2022.

**Abstract:** All new grain bins produced after 2018 are recommended to have anchor points capable of handling a 2000 lb loading for attachment of bin entry lifeline systems. This study aims to assess the feasibility of a safe entry anchor point retrofit by using finite element analysis (FEA). We used a grain bin owned by Penn State for 3D FEA modeling in SolidWorks. To validate the model results from the FEA model, first strain and then deflection measurements were conducted on the grain. Strain gauges were applied to the grain bin in five locations and strain values were obtained after applying static loads. The strain gauge measurements from the experimental study were compared to the strain output from the FEA simulation. The error seen was far greater than was expected. The most pertinent error source was strain gauge installation error and equipment failure. Then, the vertical roof deflection of the bin was measured using a precision phase-comparison laser while applying incremental static loads to the retrofitted rescue anchor points. The FEA model results were compared to the experimentally measured deflection results. A 3D FEA model of a grain bin was created. A high amount of error was observed in deflections between the measured and FEA modeling. The errors have resulted from the assumptions made during the model creation. However, the SolidWorks Simulation model still may be used to estimate loading scenarios in a safe and non-destructive way. Based on the research findings, the project team recommends that the suitability of any bin to safely accommodate a lifeline and anchor point system must be verified on a case-by-case basis. Evaluation by a professional structural engineer and consulting with the manufacturer are recommended. This recommendation extends to all-grain bins, including those post-2018.

**Keywords:** deflection; finite element analysis (FEA); grain bin; retrofit

## 1. Introduction

As defined by Occupational Safety and Health Administration (OSHA) 1910.146(b) standard, grain bins are a type of confined space that consists of limited/restricted means of entry or exit and be so configured that an employee can enter and complete required work. They are not designed for continuous human occupancy [1]. Hazards associated with grain bins include falls, exposure to dust, dust explosions, hazardous gases, entrapments, engulfment, electrical hazards, and entanglements [2] (pp. 3–4) [3] (pp. 159–169) [4]. OSHA's Grain Handling Facilities Standard (1910.272) and Permit-Required Confined Spaces Standard (1910.246) contain requirements for controlling hazards associated with grain storage structures. OSHA's standards were credited with the reduction in the number of accidents/injuries at non-exempt grain facilities [5] (pp. 1–18).

Farmers enter grain bins for various reasons but mostly to dislodge the material [6] (pp. 228–229). According to the Purdue Agricultural Confined Space Incident Database

(PACSID), 1731-grain storage and handling-related incidents were reported between 1962 and 2020 in the United States, resulting in an injury, fatality, or required emergency extrication by first responders [7] (pp. 1–19). In 2020, 15 fatalities and 20 non-fatal grain entrapment-related injuries were reported.

Grain entrapments in grain storage structures occur due to flowing grain, grain avalanche, bridging, or vacuum equipment [8] (pp. 59–72) [9] (pp. 123–134). OSHA Standard 29 CFR 1910.272 issues safety guidelines regarding grain bin entry for workers. This standard highlights the use of harnesses and lifelines in grain storage facilities. Employers should provide body harnesses with a lifeline for employees whenever they need to enter a grain storage structure or whenever they need to walk in or stand on stored grain. These lifeline systems must be engineered to support the forces imparted on them during an entrapment incident. The 29 CFR 1910.272 standard also emphasizes issuing permit procedures for entering bins, silos, or tanks. However, farming operations that employ ten or fewer employees and do not maintain temporary labor camps are exempt from OSHA enforcement even though this standard does apply to small farm operations.

American Society of Agricultural and Biological Engineers (ASABE) standard ANSI/ASABE S624 recommends that grain bins manufactured after 2018 be built to accommodate a bin entry lifeline system. The recommended system should include two anchor attachment points, one located near the roof peak and one near the roof access. Anchor attachment points must support a minimum ultimate load of 2000 pounds [10]. Newer grain bins are required to be equipped with anchor points rated to support these forces, but it is unknown if preexisting grain bins possess the structural integrity to handle these forces. The efficacy of a lifeline as a safety intervention for on-farm grain bins needs to be systemically evaluated.

Recently manufactured grain bins have engineered safe entry anchor points, but it is unclear whether their load rating is consistent among manufacturers and whether the load rating complies with the newest standards. Performing an engineering assessment of these on-farm bins will help determine retrofit options for safe anchor points. Such a study can also determine if existing on-farm bins can withstand the forces imparted on them during an impact event from someone falling or from an event where someone is being drawn down by flowing grain. The development of an engineering modeling tool will help ensure anchor point design and modifications that will not present additional safety hazards. This will give design engineers an innovative tool that will aid in giving many farms across the U.S. the option of safer bin entry. Retrofitting a bin with anchor points can be a very cost-effective option with minimal cost to the owner, compared to purchasing a new bin with the technology already built into them.

The purpose of this study was to evaluate the characteristics of existing on-farm grain storage structures and the feasibility of a safe entry anchor point retrofit for a grain bin. The specific objectives comprised developing a 3D model of a grain bin to assess the feasibility of safe entry anchor point retrofit, creating inspection criteria, and providing recommendations for safe entry retrofit. A 3D modeling solution was created to assist engineers in safely assessing on-farm grain bin lifeline retrofit potential.

## 2. Materials and Methods

The overall flow diagram for this study is shown in Figure 1. Physical strain and deflection measurements were conducted separately.

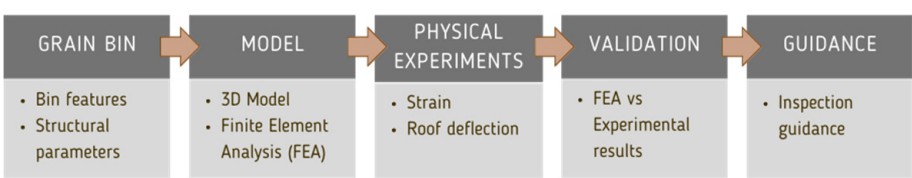

**Figure 1.** Flow diagram.

### 2.1. Bin Specifications

A grain bin owned by Penn State's Farm Operations and Services was used for modeling and deflection experiments. The bin specifications are given in Table 1.

**Table 1.** Bin specifications.

| Parameter | Values |
|---|---|
| Radius (mm) | 3657.6 |
| Roof angle (deg) | 30 |
| Fill collar radius (mm) | 401.6 |
| Fill collar thickness (mm) | 3.8 |
| Roof thickness (mm) | 0.76 |
| Fill collar lip height (mm) | 50.8 |
| Fill collar flange length (mm) | 76.2 |
| The corrugation of bin sheets (mm) | 101.6 |
| Bin sheet height (mm) | 1117.6 |
| Bin sidewall height (mm) | 5588.0 |
| Bin sheet 1 thickness * (mm) | 2.0 |
| Bin sheet 2 thickness (mm) | 1.8 |
| Bin sheet 3 thickness (mm) | 1.4 |
| Bin sheet 4 thickness (mm) | 1.3 |
| Bin sheet 5 thickness (mm) | 1.0 |

* denotes bottom bin sheet.

### 2.2. Modeling of the Grain Bin

To determine the feasibility of retrofits for safety anchor points, a 3D computer model of a grain bin was created based on the parameters from ANSI/ASABES624 standard. A surface-based 3D grain bin model was created in SolidWorks (Version 2022EDU, Dassault Systèmes SolidWorks Corporation, Waltham, MA, USA). When modeling the grain bin, the following parameters were considered: bin sheet thicknesses, roof angle, roof thickness, diameter, fill hole opening diameter, and fill hole material thickness with the following assumptions:

- The bin was in excellent condition and did not have any oxidation or structural deficiencies.
- The grain bin roof entry hatch is not a structurally significant piece to model because of the large amount of variation in entry hatch designs and sizes.
- The ladders on the grain bins offer no significant structural support in terms of dissipation of deflection.
- Bin roof stiffening rings were left out because they added additional complexity to the model.
- Weather or temperature conditions were not considered in the model.
- There were no aftermarket or user-manufactured changes or equipment added to the grain bins.
- The 3D model assumed no vertical or horizontal seams in the bin walls or in the bin roof.
- The thicknesses were applied to each face instead of having hard corners.
- There was no grain in the bin.

Rather than a solid body, surface modeling and bonding allowed for better interfacing between parts. The model began as a sketch of a single corrugation and the corrugation was then revolved 360° to make a ring, which was then patterned eleven times upon itself to make one full bin sheet ring. Instead of making separate sheets and bonding them together, it was assumed that the rings were solid without seams. This was assumed due to the rings overlapping (where one ring ends and the next begins, the first corrugation of the new ring and the old ring are overlapped).

Because it is known that these grain bins are structurally sound and can hold grain when filled, it was confidently assumed that the overlap between sheets was not going to be a point of failure during the testing. This same thought process was applied to the roof of the grain bin as well, which was bonded to the edge of the top sheet in the model (Figure 2).

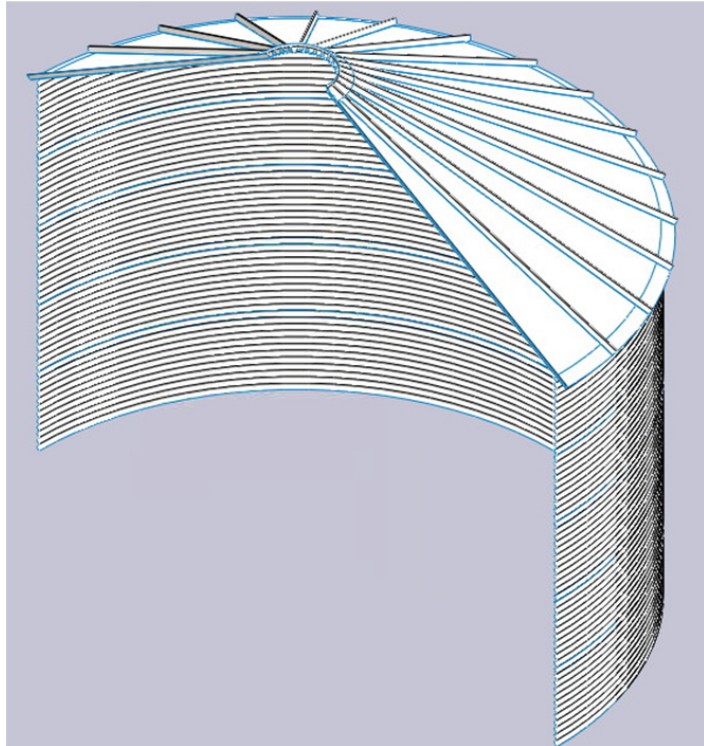

**Figure 2.** Cutaway of completed bin model.

Finite Element Analysis was used in SolidWorks to perform static analysis for loadings. During the linear FEA process (elastic stage), the finite-element model (FEM) of the grain bin was constructed using 140,564 elements. Galvanized steel with an elastic modulus of 193,105 MPa (28,007,547 psi) was used in the model. This elastic modulus was part of the software materials database and was indicated for this experiment because the grain bin being tested was made of galvanized steel.

### 2.3. Strain Measurement

The first stage of this study was constructed by experimenting with strain measurements on the grain bin to simulate an entrapment incident, which would be measured through the loading of the retrofitted safe entry anchor points. The strain seen at certain points on the grain bin would be correlated to the corresponding locations on the 3D model of the grain bin, subjected to the same simulated loadings.

Before beginning with the strain gauge installation, safe entry anchor points were retrofitted to the grain bins. A 10 k zinc-plated steel swivel anchor (Guardian Fall Protection MEGA Swivel, Guardian, LLC, Kent, WA, USA) was installed on the peak collar on both bins and a sidewall anchor (#1550 D-Bolt, FrenchCreek Fall Safety, Franklin, PA, USA) was installed directly under the manhole entrance at the bottom of the roof on both bins. The sidewall d-ring anchors were mounted to the sidewall inside of the top three full corrugations, and the peak swivel anchors were mounted inside the peak collar in line with the manhole. Strain gauges (C2A-06-250LW-350, Micro-Measurements, Raleigh, NC, USA) were then applied to the grain bin in five locations as seen in Figure 3. Each location had two strain gauges, mounted perpendicularly from each other to measure both axial and transverse strain.

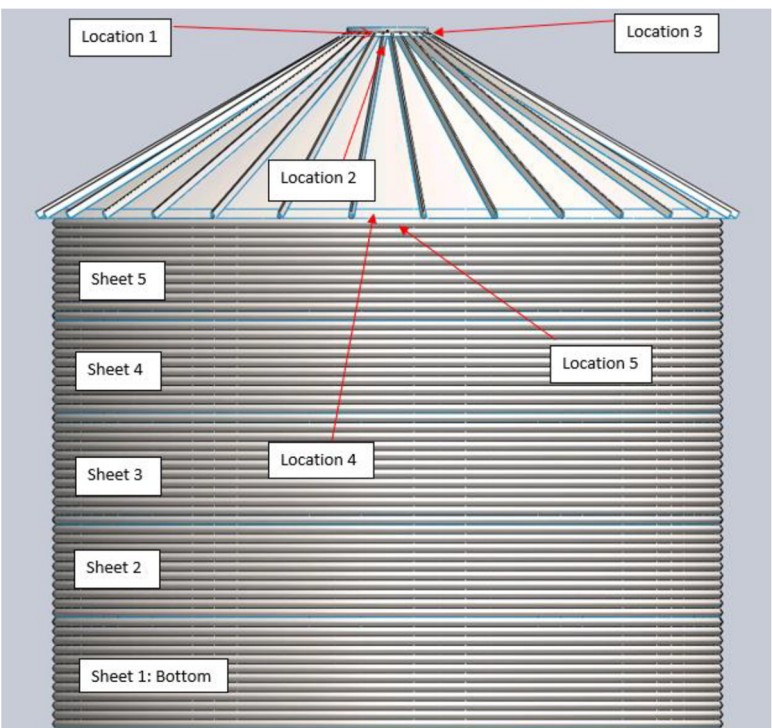

**Figure 3.** Perpendicular arrangement of strain gauge pairs and measurement locations.

A temporary loading cable was tied off at the sidewall rescue anchor point and was routed up over a pulley connected to the roof peak anchor point. The cable was loaded by adding tractor suitcase weights. The loading ranged from 7.7 kg (the weight of the tackle without suitcase weights) to a maximum of 297.1 kg. The load was measured by a digital crane scale with an accuracy of 0.5 kg.

*2.4. Deflection Measurements*

Even though the choice to use strain gauges theoretically allowed for a very in-depth understanding of how the grain bin reacted to forces, it turned out to be a method that was not the most feasible. So, the project team decided to use deflection measurements in the second stage of the study. Deflections, also known as displacements, can occur from external loads or from the weight of the structure itself. In order to validate the model, the vertical roof deflection of the bin was measured using a precision phase-comparison laser sensor while applying incremental dead loads to the retrofitted rescue anchor points. The laser sensor used was a Micro-Epsilon optoNCDR ILR1182-30 (Micro-Epsilon Messetechnik, Ortenburg, Germany) which has a resolution of 0.1 mm and repeatability of ≤0.5 mm from 0.1 to 50 m. The sensor was mounted on the sweep auger shield 838 mm from the grain bin center point as shown in Figure 2 and aimed at a spot on the roof directly above it. Since the sweep auger is supported by only the concrete center and circumferential footing structures and not the perforated metal floor, measurements were isolated from any floor deflections caused by the moving and loading of weights (Figure 4). Similar to the strain measurement, a temporary loading cable was used to load the weights. This time we incrementally loaded and unloaded by adding/removing ten tractor suitcase weights of approximately 29 kg one at a time to a maximum of 297.1 kg.

*2.5. Inspection Guidance*

Taking and packaging the information from this study into a tool that end users could take and implement on their farms was part of our deliverables. This delivery took shape as an inspection guidance document. This guidance will give farmers a tool that could be worked on with a structural engineer to determine whether their grain bin is a viable

candidate for a safe entry lifeline retrofit. When considering the important parts of a grain bin to inspect, the team combined experiences from years of teaching grain bin rescue, insurance inspector considerations, and concrete inspection methods.

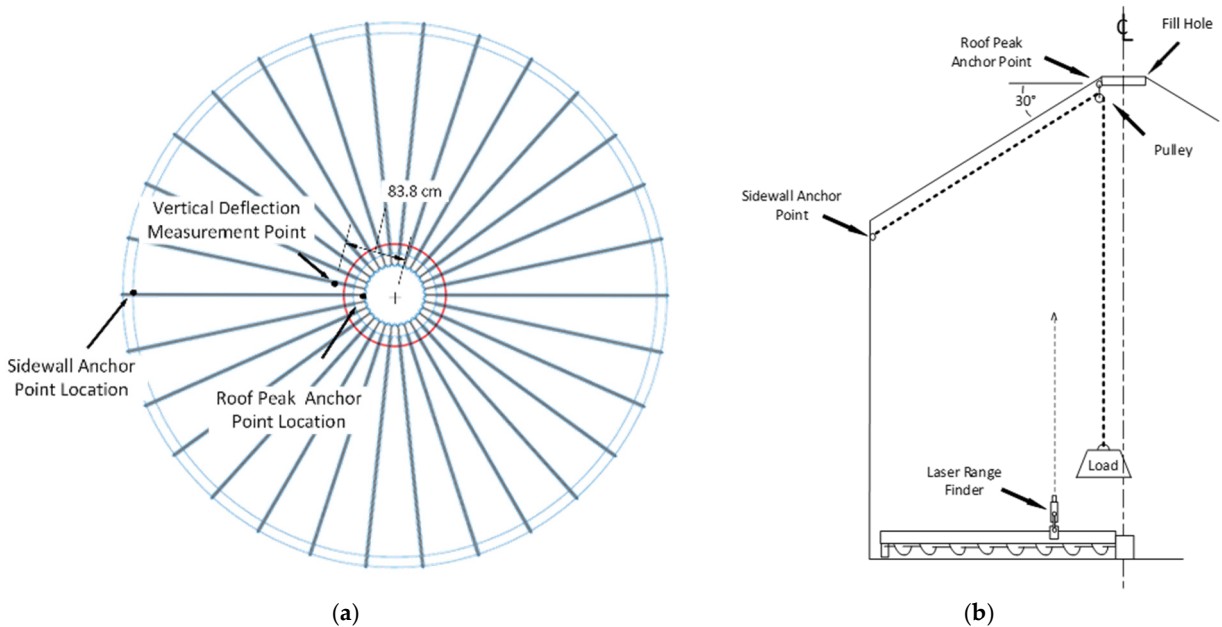

| (a) | (b) |

**Figure 4.** Location of deflection measurement: (**a**) plan view (**b**) side view/sketch of corrugations.

## 3. Results

*3.1. Strain Measurement Results*

Strain data collection was conducted using the methods and recorded by hand in a notebook. That data was then transferred into Microsoft Excel for processing. The three individual test results for each bin were averaged together (Table 2). Strain0 ($\varepsilon_0$) refers to the average initial strain measurement after "zeroing" the test equipment and Strain1 ($\varepsilon_1$) refers to the average strain measurement after the loadings. Strain values were shown in units of microstrain.

**Table 2.** Experimental strain.

| | Average | | | | |
|---|---|---|---|---|---|
| **Location** | **Strain0 ($\varepsilon_0$)** | **Strain1 ($\varepsilon_1$)** | **Diff ($\Delta\varepsilon$)** | **Min** | **Max** |
| 1_y | $-1.00 \times 10^{-6}$ | $-1.47 \times 10^{-5}$ | $-1.37 \times 10^{-5}$ | $-2.00 \times 10^{-5}$ | $-8.00 \times 10^{-6}$ |
| 1_x | $2.33 \times 10^{-6}$ | $-1.60 \times 10^{-5}$ | $-1.83 \times 10^{-5}$ | $-2.00 \times 10^{-5}$ | $-1.50 \times 10^{-5}$ |
| 2_y | $-6.67 \times 10^{-7}$ | $5.63 \times 10^{-5}$ | $5.70 \times 10^{-5}$ | $4.20 \times 10^{-5}$ | $7.10 \times 10^{-5}$ |
| 2_x | $0.00 \times 10^{0}$ | $1.03 \times 10^{-5}$ | $1.03 \times 10^{-5}$ | $6.00 \times 10^{-6}$ | $1.50 \times 10^{-5}$ |
| 3_y | $0.00 \times 10^{0}$ | $-2.37 \times 10^{-5}$ | $-2.37 \times 10^{-5}$ | $-3.30 \times 10^{-5}$ | $-1.40 \times 10^{-5}$ |
| 3_x | $2.00 \times 10^{-6}$ | $-8.33 \times 10^{-6}$ | $-1.03 \times 10^{-5}$ | $-1.20 \times 10^{-5}$ | $-8.00 \times 10^{-6}$ |
| 4_y | $6.67 \times 10^{-7}$ | $-5.26 \times 10^{-4}$ | $-5.27 \times 10^{-4}$ | $-5.33 \times 10^{-4}$ | $-5.20 \times 10^{-4}$ |
| 4_x | $1.00 \times 10^{-6}$ | $6.50 \times 10^{-5}$ | $6.40 \times 10^{-5}$ | $5.70 \times 10^{-5}$ | $7.40 \times 10^{-5}$ |
| 5_y | $3.33 \times 10^{-7}$ | $9.90 \times 10^{-5}$ | $9.87 \times 10^{-5}$ | $9.70 \times 10^{-5}$ | $1.01 \times 10^{-4}$ |
| 5_x | $-6.67 \times 10^{-7}$ | $1.50 \times 10^{-4}$ | $1.50 \times 10^{-4}$ | $1.48 \times 10^{-4}$ | $1.53 \times 10^{-4}$ |

Upon analysis of the data, it was found that the original data set was inconclusive and further testing was to be required. The strain that was recorded from the experimental study was compared to the strain output from the FEA simulation. The error seen was far greater than was anticipated and that was found to be from a multitude of error sources, the most pertinent of which was due to installation error and equipment failure.

Using old data interpreting equipment added errors to the system. Coupled with the 3D modeling assumptions stated below, the error was too great to be conclusive, maxing out at 1109% (Figure 5).

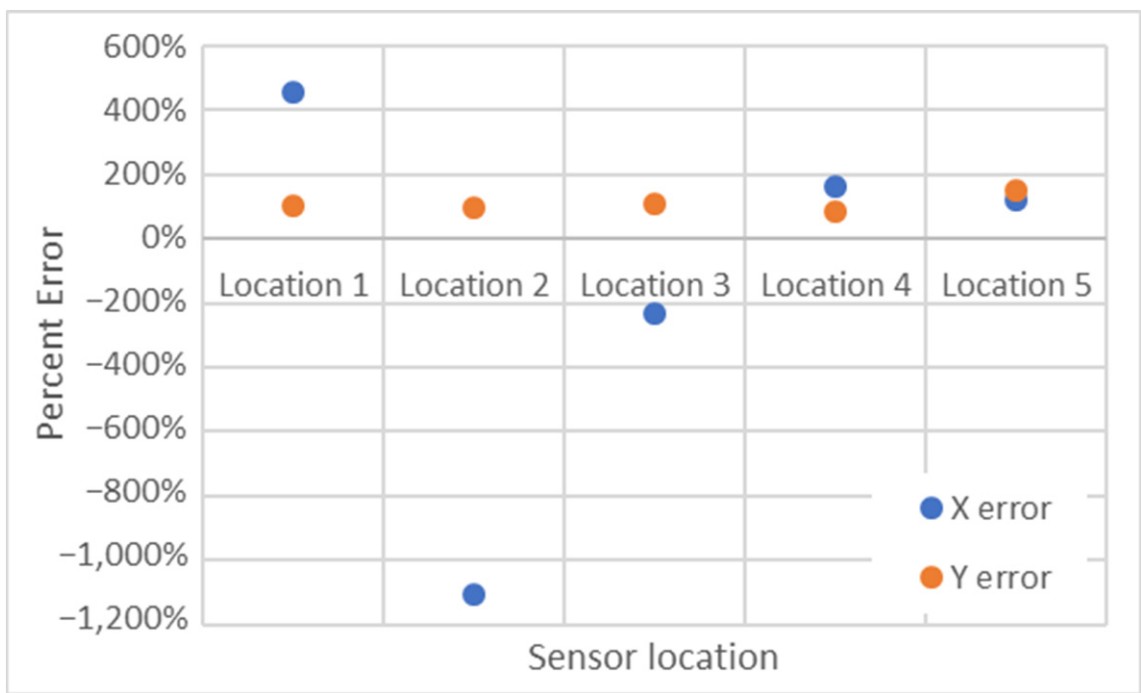

**Figure 5.** Grain bin strain error comparison.

### 3.2. Experimental Deflection Measurements

The research team used deflection measurements after the inconclusive strain results. The grain bin deflection measurements were taken after two complete loading/unloading cycles of the safe entry anchor points were performed. During these two loading/unloading cycles, the bin structure settled as the panels shifted slightly against the fasteners and one another. Data was collected during the third loading/unloading cycle. The maximum applied load on the physical grain bin was 297.1 kg (655 lb). Testing was performed such that no damage to the bin would occur, so the 2000 lb load recommended by ANSI/ASABE S624 was not tested experimentally. The deflection at the rooftop anchor point varied between 0 and 2.4 mm. The deflections were slightly higher for unloading conditions than those measured during the loadings.

### 3.3. Finite Element Analysis (FEA) Results

Displacement under each loading was obtained from the FEA model and presented as heat diagrams (Figures 6 and 7) The maximum load applied on the physical grain bin was 297.1 kg (655 lb). The 907.2 kg (2000 lb) load was not applied due to the age of the bin and the need to ensure that any physical testing on the bin will not cause any permanent and irreversible damage to the bin. The FEA model for the 907.2 kg (2000 lb) was chosen to model for the minimum ultimate load supported by anchor attachment points stated in the ANSI/ASABE S624 Grain Bin Access Design Safety Standard. It should be noted that the anchor point is located on the left side of the grain bin in the figures. As expected, higher loadings resulted in larger predicted deflections in the FEA model. At the vertical defection measurement point shown in Figure 6a, the model predicted 5.3 mm of vertical deflection for anchor loadings of 297.1 kg (655 lb) while it was 16 mm for the maximum anchor loadings of 907.2 kg (2000 lb). As shown in Figure 7, a nominal amount of deflection is noted at the side anchor point.

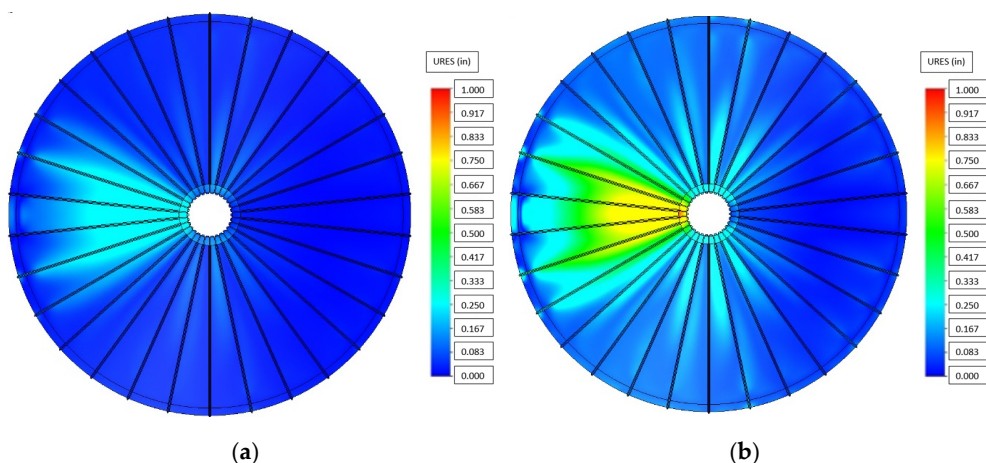

**Figure 6.** Maximum displacement on the bin rooftop in the FEA model as induced by the anchor loadings (top view): (**a**) 297.l kg (655 lb), (**b**) 907.2 kg (2000 lb).

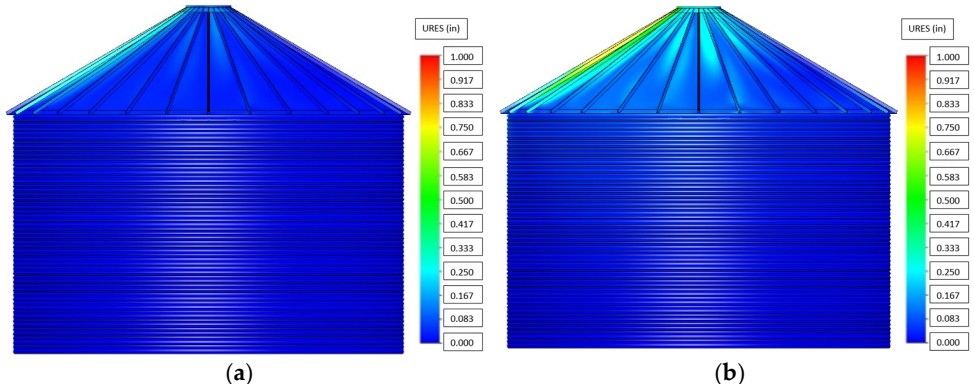

**Figure 7.** Maximum displacement on the bin in the FEA model as induced by the anchor loadings (side view): (**a**) 297.l kg (655 lb), (**b**) 907.2 kg (2000 lb).

### 3.4. Model Validation

The measured and predicted deflections by the finite element analysis for various loading and unloading conditions are shown in Table 3. The deflection values during loading and unloading experiments aligned with the FEA-predicted deflections. Even though the deflections were within the same order of magnitude ranging between 100% and 250%, the model significantly overestimated the deflections.

**Table 3.** Experimental and model results comparisons.

| Loading—Deflection (mm) | | | Unloading—Deflection (mm) | | |
|---|---|---|---|---|---|
| Load (kg) | Measured | FEA | Load (kg) | Measured | FEA |
| 7.7 | 0.0 | 0.1 | 297.1 | 2.4 | 4.5 |
| 36.3 | 0.2 | 0.5 | 269.0 | 2.2 | 4.0 |
| 65.3 | 0.4 | 1.0 | 240.0 | 2.0 | 3.6 |
| 93.9 | 0.6 | 1.4 | 210.9 | 1.8 | 3.6 |
| 122.9 | 0.9 | 1.8 | 181.4 | 1.6 | 2.7 |
| 152.0 | 1.2 | 2.3 | 152.9 | 1.3 | 2.3 |
| 181.4 | 1.4 | 3.7 | 123.4 | 1.0 | 1.8 |
| 210.9 | 1.6 | 3.2 | 94.3 | 0.8 | 1.4 |
| 240.0 | 2.0 | 3.6 | 65.8 | 0.5 | 1.0 |
| 269.4 | 2.1 | 4.0 | 36.7 | 0.2 | 0.8 |
| 297.1 | 2.4 | 4.5 | 7.7 | 0.0 | 0.1 |

*3.5. Grain Bin Inspection Guidance for Safe Entry Lifeline and Anchor Point Retrofit*

Inspection guidance was developed for structural engineers to assess grain bins and the feasibility of retrofitting safe entry anchor points onto on-farm, non-stiffened grain bins. The structure of the inspection guidance was similar to the FARM-HAT (Farm/Agriculture/Rural Management- Hazard Analysis Tool) [11]. FARM-HAT is a hazard analysis tool for evaluating hazards, and recommendations on correcting hazards and it can be used for farming, ranching, agri-business, etc., related hazards (Penn State Extension, 2022). FARM-HAT which has been renamed SaferFarm and is available as a mobile-device friendly website that was developed and now maintained by the National Farm Medicine Center and the Marshfield Clinic Research Foundation, Marshfield, WI, USA.

The guidance breaks down the bin sections to allow for careful inspection of each parameter of the grain bin. These parameters were to include the concrete foundation, the connection to the concrete foundation, the caulking, the sidewall, the hardware, and the grain bin roof. By carefully inspecting these components, an engineer may determine the feasibility of a safe entry lifeline retrofit. Each section can be used for examining and rating the conditions from poor to excellent. The inspection criteria are available at Penn State Extension (Figure 8) (https://extension.psu.edu/on-farm-grain-bin-inspection-guidance-for-safe-entry-lifeline-and-anchor-point-retrofit accessed on 1 December 2022).

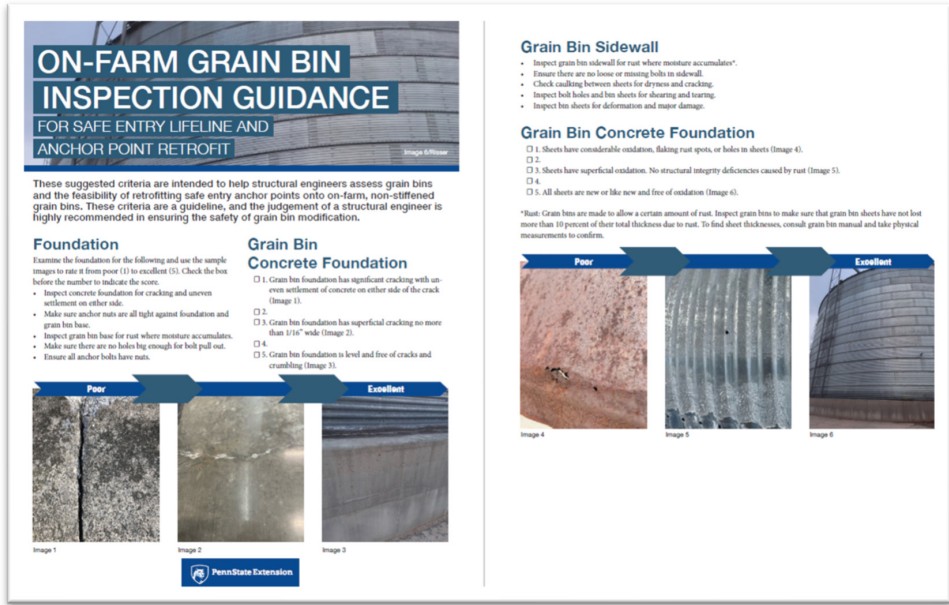

**Figure 8.** Preview of the on-farm grain bin inspection guidance.

## 4. Discussion

When entering a grain bin, all entrants should wear a suitable harness connected to a lifeline with anchor support; however, not all grain bins have the anchor and lifeline system. In this study, a finite element analysis model was developed for a grain bin to simulate the conditions of an anchor point for safe entry to a grain bin. Initially, we used strain measurement to simulate the anchor loading conditions, however, the results were inconclusive. The research team decided to transition to a deflection measurement method. The experimental deflection results and FEA model results were within the same order of magnitude, even though the amount of error was very high in percentage. One explanation for the higher amount of difference would be that there was shifting in the panels and fasteners when loading and unloading for the actual measurement. The shifting on the panels was not measured and the transfer load to the side wall may reduce the overall roof deflection. Additionally, the model did not account for weathering, degradation,

modifications, and exact conditions of the grain bin and these conditions would also increase the amount of deflection.

The goal of this research was to continue research and outreach in order to make on-farm grain storage safer. We recognize that there were a number of gaps in our research. Utilizing strain measurement techniques, the maximum error seen was around −1100% error, which was due to human factors, not the measurement method itself. By changing physical assessment methods from strain to deflection measurement we were able to decrease error percentages by almost 900%. This is a step in the right direction, but even 250% error is still far too much error to be conclusive when it comes to the physical evaluation of a grain bin for safety purposes.

Through data collection and modeling, it was determined that there are too many variations among grain bins. Because of various types and modifications, grain bins must be assessed on a case-by-case basis whether a retrofit is feasible. In our study, we only modeled one size grain bin; therefore, further research is needed to understand bins of other sizes. It might be possible to improve the model by including additional variables such as different size bins, loading and unloading conditions, and the effect of shifting side panels/walls during loading/unloading.

Grain bin owners can check with their grain bin dealer or manufacturer, or they can work with a structural engineer to determine the feasibility of retrofitting. These criteria are a guideline, and the judgment of a structural engineer is highly recommended in ensuring the safety of grain bin modifications. The inspection criteria are intended to be a starting point to quantitatively assign values to the physical condition of a grain bin. These values are not a definitive answer nor do these values necessarily mean that your grain bin would be able to be successfully retrofitted with anchors points and a lifeline but should be used as a guide in the decision to retrofit the grain bin. This tool is to be utilized by structural engineers to assist in the decision-making process.

## 5. Conclusions

In conclusion, we concluded that finite element analysis can be vital to assess safety-critical structures such as grain bins. FEA can be used by all engineers to assess the structural properties of grain bins before installing an anchor point. Further studies should consider the following conclusions and recommendations:

- Because of the varying size, environmental and working conditions, and modifications, the suitability of any bin to safely accommodate a lifeline and anchor point system must be verified on a case-by-case basis. Future research is needed to verify FEA models for different sizes of bins.
- Evaluation by a professional structural engineer and consulting with the manufacturer are recommended. This recommendation extends to all-grain bins, including those post-2018.
- Engineering evaluation for safely accommodating a lifeline and anchor point system is essential for bins that are modified, damaged, or have other signs of degradation.
- Educational efforts focusing on technologies and best practices can reduce the need for grain bin entry.
- Our model excluded bin roof stiffeners in the model. The FEA model accuracy might be improved by including roof stiffeners and load paths between roof-to-wall connections.
- In the FEA model, even though there is an anchor point on the side of the bin, the model showed no deflections on the side of the bin. The model accuracy might be improved by including transferred loads on the side wall. This transferred load might reduce the overall deflection on the roof.
- The amount of shifting in the panels and fasteners should be measured during the load/unload cycle and this might explain the difference between model and measurement values.

**Author Contributions:** Conceptualization, M.D.; methodology, M.D. and R.T.; software, R.B. and M.D.; validation, M.D. and R.B.; formal analysis, M.D. and S.G.; resources, L.F.; data curation, M.D.; writing—original draft preparation, M.D. and S.G.; writing—review and editing, L.F.; visualization, M.D., R.T. and R.B.; supervision, S.G. and J.L.; project administration, L.F. All authors have read and agreed to the published version of the manuscript.

**Funding:** This work was supported by a grant from the Northeast Center for Occupational Health and Safety: Agriculture, Forestry, and Fishing (NIOSH grant #2U54OH007542). This work was supported in-part by the USDA National Institute of Food and Agriculture Federal Appropriations under Project PEN04671 and Accession number 1017582.

**Institutional Review Board Statement:** Not applicable.

**Informed Consent Statement:** Not applicable.

**Data Availability Statement:** The data are not publicly available.

**Acknowledgments:** The authors wish to acknowledge the following individuals and institutions for their contributions to this research: Stephen Brown, former Penn State Extension Associate for helping with the experiments, and Penn State Farm Operations for the use of grain bins.

**Conflicts of Interest:** The authors declare no conflict of interest.

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
