# Peer review of "Case Study: Modeling a Grain Bin for Safe Entry Retrofitâ€"

_safety_

Round 1
Reviewer 1 Report
The purpose of this study was to evaluate the characteristics of existing on-farm grain storage structures and the feasibility of a safe entry anchor point retrofit for a grain bin. The specific objectives comprised developing a 3D model of a grain bin to assess the feasibility of safe entry anchor point retrofit, creating inspection criteria, and providing recommendations for safe entry retrofit. To determine the feasibility of retrofits for safety anchor points, a 3D computer model 99 of a grain bin was created based on the parameters from ANSI/ASABES624 standard. The authors concluded that finite element analysis can be vital to assess safety critical structures such as grain bins. FEA can be used by all engineers to assess the structural properties of grain bins before installing an anchor point. Through data collection and modeling, it was determined that there are too many variations among grain bins. Because of various types and modifications, grain bins must be assessed on a case-by-case basis whether a retrofit is feasible.
The paper is well done I have not remarks
Reviewer 2 Report
This manuscript is well written and it presents important information for farmers and the grain trade about a very dangerous situation which can occur when grain is flowing inside a bin. I would like to have seen larger diameter bins tested. A 24 ft diam bin is quite small by today's standards.
Author Response
Thank you for the review. The conclusion was updated, and those changes are marked and visible in the revised manuscript. Additional text was added in the Discussion section (lines 280 and 281). A section in Discussion was edited and moved to the Conclusion section (lines 334 – 356). Please see the attachment.

Reviewer 3 Report
The abstract does not provide the reader with information about the results. It has only one numeric value of the results. It needs to be improved, giving more numeric values for the results.
In my respect of view, the authors have made a lot of assumptions in order to model the grain bin. Did they try to investigate how each of these assumptions affects the efficiency of the finite-element model? My opinion is that some of them, such as the weather temperature condition and the absence of vertical or horizontal seams in the bin walls or in the bin roof affect the efficiency of the FEM.
Please add a separate Conclusion section. The Conclusions should give useful information about this work. Maybe the authors should include some numerical values.
Author Response

(The authors gave the same response as above.)

Round 2
Reviewer 1 Report
The authors have applied the review' s suggestions
Reviewer 3 Report
My comments have been addressed.